# Antibiotic Resistance of *Escherichia coli* Isolated from *Conventional*, *No Antibiotics*, and *Humane Family Owned* Retail Broiler Chicken Meat

**DOI:** 10.3390/ani10122217

**Published:** 2020-11-26

**Authors:** Helen M. Sanchez, Victoria A. Whitener, Vanessa Thulsiraj, Alicia Amundson, Carolyn Collins, Mckenzie Duran-Gonzalez, Edwin Giragossian, Allison Hornstra, Sarah Kamel, Andrea Maben, Amelia Reynolds, Elizabeth Roswell, Benjamin Schmidt, Lauren Sevigny, Cindy Xiong, Jennifer A. Jay

**Affiliations:** 1Department of Civil and Environmental Engineering, University of California at Los Angeles, Los Angeles, CA 90095, USA; Helen.M.Sanchez@usace.army.mil (H.M.S.); victoria.whitener@engineering.ucla.edu (V.A.W.); vthulsiraj@ucla.edu (V.T.); ara121094@hotmail.com (A.A.); mckenzieduran2019@gmail.com (M.D.-G.); ahreynolds@ucdavis.edu (A.R.); 2Institute of the Environment and Sustainability, University of California at Los Angeles, Los Angeles, CA 90095, USA; caricollins345@gmail.com (C.C.); edwin_giragossian@yahoo.com (E.G.); avhornstra@gmail.com (A.H.); skamel@ucla.edu (S.K.); andrea.maben@gmail.com (A.M.); earoswell@gmail.com (E.R.); benjamin.elliot10@gmail.com (B.S.); laurensevigny@gmail.com (L.S.); cxiong@u.northwestern.edu (C.X.)

**Keywords:** antibiotic resistance, antimicrobial resistance, poultry, chicken, ampicillin, erythromycin, *Escherichia coli*, food reservoir, retail meat, One Health

## Abstract

**Simple Summary:**

While it is well known that antibiotics administered for either therapeutic or non-therapeutic purposes in livestock farms promote the development of antibiotic resistance in bacteria through selective pressure, there are conflicting findings in the literature with regard to the influence of production strategies on antibiotic resistance in bacteria isolated from commercially-available chicken. In this work, we tested the hypothesis that there would be differences in antibiotic resistance in *E. coli* isolated from three categories of production methods: *Conventional*, *No Antibiotics*, and *Humane Family Owned*. In this work, it was found that for both ampicillin and erythromycin, there was no significant difference (*p* > 0.05) between *Conventional* and USDA-certified *No Antibiotics* chicken, which is in line with some previous work. The novel finding in this work is that we observed a statistically significant difference between both of the previously mentioned groups and chicken from *Humane Family Owned* production schemes. To our knowledge, this is the first time *E. coli* from *Humane Family Owned* chicken has been studied for antibiotic resistance. This work contributes to a better understanding of a potential strategy of chicken production for the overall benefit of human health, in line with the One Health approach implemented by the World Health Organization.

**Abstract:**

The use of antibiotics for therapeutic and especially non-therapeutic purposes in livestock farms promotes the development of antibiotic resistance in previously susceptible bacteria through selective pressure. In this work, we examined *E. coli* isolates using the standard Kirby-Bauer disk diffusion susceptibility protocol and the CLSI standards. Companies selling retail chicken products in Los Angeles, California were grouped into three production groupings—*Conventional*, *No Antibiotics*, and *Humane Family Owned*. *Humane Family Owned* is not a federally regulated category in the United States, but shows the reader that the chicken is incubated, hatched, raised, slaughtered, and packaged by one party, ensuring that the use of antibiotics in the entire production of the chicken is known and understood. We then examined the antibiotic resistance of the *E. coli* isolates (*n* = 325) by exposing them to seven common antibiotics, and resistance was seen to two of the antibiotics, ampicillin and erythromycin. As has been shown previously, it was found that for both ampicillin and erythromycin, there was no significant difference (*p* > 0.05) between *Conventional* and USDA (United States Department of Agriculture)-certified *No Antibiotics* chicken. Unique to this work, we additionally found that *Humane Family Owned* chicken had fewer (*p* ≤ 0.05) antibiotic-resistant *E. coli* isolates than both of the previous. Although not considered *directly* clinically relevant, we chose to test erythromycin because of its ecological significance to the environmental antibiotic resistome, which is not generally done. To our knowledge, *Humane Family Owned* consumer chicken has not previously been studied for its antibiotic resistance. This work contributes to a better understanding of a potential strategy of chicken production for the overall benefit of human health, giving evidentiary support to the One Health approach implemented by the World Health Organization.

## 1. Introduction

Antibiotic resistance is a growing global health concern with the huge societal risk of reverting to a pre-antibiotic era if not addressed. There is a well-documented connection between the rise in antibiotic-resistant infections and the ninety billion tons of chicken meat that are produced worldwide annually [1,2]. By 2050, it is estimated that there will be a death resulting from an antibiotic-resistant infection every three seconds [3]. Because of this connection, the World Health Organization (WHO) adopted the One Health approach in 2017, which states that the health of people, animals, and the environment are inextricably related to one another. Expanding upon the topic, the WHO has stated that the One Health approach is particularly relevant when discussing antibiotic resistance, food safety, and the control of zoonoses. An abundance of previous work has found that antibiotic use in animals, especially when it is sub-therapeutic, impacts the microbiome of the animal [4,5,6,7], which then affects the resistome in the surrounding environment [8,9,10]. Infections caused by antibiotic-resistant bacteria result in an increased frequency of hospitalization for a person as well as longer hospital stays than for susceptible (not antibiotic resistant) infections [11,12,13].

It was over forty years ago that the Food and Drug Administration (FDA) recognized that antibiotic resistance poses a human health concern when antibiotics are given at sub-therapeutic doses to animals [14]. Recently in 2017, the FDA implemented new rules on the use of clinically relevant antibiotics in food-animal production. Clinically relevant antibiotics are those that are used for the treatment of diseases in humans. With these new rules, clinically relevant antibiotics are no longer allowed as growth promoters or for feed efficiency in chickens and other food animals. While this is a big improvement in regulation, antibiotics considered “not medically relevant” are still allowed in chicken feed.

The problem is that resistance genes are generally co-located on plasmids with other antibiotic resistance genes, and while resistance to “medically important” antibiotics may not be triggered by a “not medically important” antibiotic, the genes that we do think of as “medically important” will be able to propagate as a result of the use of these “not medically important” antibiotics. Additionally, there is a growing body of research on the particular affinity of antibiotic resistance genes to transfer horizontally in nature. As will be discussed further below, it cannot be assumed that antibiotic resistance infections in humans will be effectively minimized by only reducing clinically relevant antibiotics used in food animal production because of the ability of these genes to transfer from commensal bacteria to pathogenic bacteria. As a result of complex biological interactions such as these, the Centers for Disease Control and Prevention (CDC) announced in their recent 2019 report that one of the top priorities in stopping the threat of antibiotic resistance needs to be a better “understanding of the role of antibiotic-resistant [bacteria, fungi, and viruses] in the environment” [15].

While much attention has been focused on the clinical overuse of antibiotics, up to 70 percent of antibiotics produced in the United States in 2014 were sold for use in livestock alone [16]. Agricultural antibiotic standards from the USDA have changed since 2014, and the agricultural use of “medically important” antibiotics decreased by 36% from 2014 to 2018.

However, the use of what are considered “not medically important” antibiotics has remained virtually unchanged [17]. The practice of administering sub-therapeutic doses of antibiotics to prevent disease in livestock not only increases resistance in bacteria found in animals, but also in humans [18,19,20,21,22]. The United States is currently the second highest user of antibiotics in food animal production. With regard to agricultural use, the WHO’s Antimicrobial Resistance report states that “major gaps exist in surveillance and data sharing related to the emergence of ABR (antibacterial resistance) in foodborne bacteria and its potential impact on both animal and human health”. Priority next steps include development of new tools and increased surveillance in food-producing animals and the food chain [23].

Both workers and consumers may be exposed to antibiotic-resistant bacteria as a result of farming practices. Poultry farm workers and families living on farms using antibiotics in the feed, as well as the neighboring families, have an elevated risk of exposure to antibiotic-resistant *E. coli* [18,22,24]. Consumers that purchase poultry products which utilize antibiotics in production may also be exposed through consumption of undercooked meat and cross-contamination from raw meat on surfaces. Once antibiotic-resistant bacteria become a part of the human gut environment, it is known that antibiotic resistance genes are able to move horizontally, especially through conjugative gene transfer, to other bacteria [25]. *E. coli*, specifically, has particular relevance to human health because it has been suggested in several studies [26,27,28,29] that *E. coli* is able to transmit from retail meat to people and ultimately be a source of urinary tract infections.

Labels on retail poultry in the United States can be a source of much confusion to the consumer. Even if the consumer works to become educated on the meaning of antibiotic use labeling, the enigmatic definitions pose a huge barrier. Consumers are often presented with a wide variety of statements on labels regarding the quality of meat and levels of antibiotic usage during poultry production. Chickens raised with sub-therapeutic doses of antibiotics in *Conventional* farming practice may still contain labels claiming “all natural” or “free range”, which imply a healthier product even though both of these statements are silent on antibiotics use [30]. For retail poultry meat, the label *Organic* as defined and certified by the USDA (United States Department of Agriculture), requires that poultry livestock are not given antibiotics or hormones after the first 24 h of life. Thus, injecting antibiotics into eggs or administering antibiotics to chicks during the first day of life are practices that are often performed and do not violate the USDA organic standard.

There are various forms of “No Antibiotic” labels in use in the United States, such as “Raised Without Antibiotics”, “No Antibiotics Administered”, “No Added Antibiotics”, “Raised Antibiotic Free”, and “No Antibiotics Ever”, all of which describe meat that has not been administered antibiotics during production, with the “No Antibiotics Ever” being slightly stricter than the others in that it also restricts antibiotic use in the egg (*in ovo*). There is a notable difference between products that have these labels along with a USDA Process Verified seal, and those that only have the label. The label itself may be used as long as the producers send proper documentation to the USDA to support the claim, but they are approved after only a one-time desk review. Because there are no formalized standards for the antibiotic use claims, the producers develop their own antibiotics standards and terminology and present them for approval to the USDA FSIS (Food Safety and Inspection Service), and application materials (exactly what the label means) are not made public. To also use the USDA Process Verified label, the USDA Agricultural Marketing Service (AMS) also conducts a one-time on-site audit of the facility and makes public the audited process points and the standards behind them [31]. This additional certification allows for companies to provide extra transparency and assurance to consumers that the process claimed was verified by either the USDA or a third party.

For our purposes, we assigned three different categories based on third party certification and statements on the retail labels: *Conventional*, *No Antibiotics* and *Humane Family Owned*. The descriptions of the three are further explained below, but briefly, the *Conventional* chicken tested was known to be treated with antibiotics sub-therapeutically at the time of the study. The *No Antibiotics* category includes brands which make reference to no antibiotic usage but may share facilities for slaughter with brands employing antibiotics. *Humane Family Owned* poultry prohibits sub-therapeutic use of antibiotics, ionophores, beta agonists, and/or sulfa drugs before and after hatching, does not allow for sharing of slaughter facilities, and is third-party certified by a corporation prioritizing the well-being of the animal, either by the Global Animal Partnership or Certified Humane standards in our work. *Humane Family Owned* indicates that the chicken is managed from birth through slaughter by the same company without sharing of facilities. While it is not a production category of poultry, it is an appropriate descriptor of the production process of these brands in our work.

*No Antibiotics*, *Conventional*, and *Humane Family Owned* meat products provide various options to consumers, some of whom will pay a higher price for their preference according to perceived health benefits [32,33], despite that exposure to antibiotic resistance from these products is not always easily discernable from labels. This extends past the individual consumer. There are school districts [14] and private sector companies [34] in the past decade who have moved towards serving what is perceived as chicken that has not been affected by antibiotics.

While the frequency of contamination with antibiotic-resistant *E. coli* in consumer chicken has been previously studied [35], our work is the first to include the class of *Humane Family Owned* that was examined in our work. Some previous studies have shown higher antibiotic resistance in *Conventional* versus “Organic” or “Antibiotic Free” poultry [11,35,36] while others have shown the converse or no difference [37,38,39]. It is important to note that human exposure to antibiotic resistance from agriculture is not limited to the food pathway. Excreted antibiotics, antibiotic resistance genes and antibiotic-resistant pathogens are released to the surrounding environment, and subsequent selective pressure, transfer of genes among bacteria, and transport can increase resistance human exposure to environmental antibiotic resistance.

The specific labels included on the retail broiler chicken packaging in this work are detailed in Table 1, with a focus on components of the humane raising and the antibiotic use in production of the chicken. Table 2 then organizes the chicken brands purchased into three categories *Conventional*, *No Antibiotics*, *and Humane Family Owned*, and also includes all brands that were listed on the package labeling. For the purposes of our work, all *Conventional* brands happened to be USDA-certified with the label “All Natural”, meaning that the chicken must be minimally processed, have no added artificial ingredients, be all vegetarian-fed, and must not include added hormones. The “All Natural” label is approved by the USDA. In our work, NA-1 and NA-2 (Figure 2) were the two separate *No Antibiotics* brands tested. In addition to a USDA “No Antibiotics Ever” label, NA-1 also had an “All Natural” label. The other brand, NA-2, was Organic, RWA, and “Free Range.” The USDA Organic seal is additionally approved by a third party, but only after 24 h of life. In order to have a USDA Organic seal, the company must raise the chicken in conditions that accommodate their behavior, feed the chicken organic feed, and not give the animal antibiotics or hormones before 24 hours of life. “Free Range” is not a USDA label, and means that the chicken has access to the outdoors.

It is important to note that while the antibiotic use in poultry labeling and production is not easily discernable from labels, and even often misleading, this is not necessarily the fault of the USDA or FDA (United States Food and Drug Administration). It is easily accessible and comprehensible via the respective websites what USDA-certified labels mean. Other labeling is the responsibility of the individual corporation, and while it is oftentimes misleading, it is not currently directly within the domain of the USDA or the FDA to regulate these other labels.

In the United States, the National Antimicrobial Resistance Monitoring System for Enteric Bacteria (NARMS) exists to better understand and document antibiotic resistance. NARMS is a multi-agency, cooperative public health surveillance system effort between the CDC, the FDA, and the USDA in the United States [40]. While this is a huge effort meant to aid in the United States’ understanding of the antibiotic resistance trends, and appropriately recognizes that there is a connection between human, environmental, and food animals and antibiotic resistance, it is severely limited by only documenting “clinically relevant” antibiotic resistance, despite the highly mobile nature of antibiotic resistance.

This study evaluates the resistance of *E. coli* isolates from retail broiler chicken to seven different antibiotics. All chicken products sampled are commonly available in Southern California markets. The retail chicken was grouped as *Conventional*, *No Antibiotics*, or *Humane Family Owned* based upon the packaging labels. Differences in the antibiotic resistance of the *E. coli* isolates between production groupings allow us to better understand the effect that production practices have on the occurrence of antibiotic resistance in the retail meat.

## 2. Materials and Methods

### 2.1. Assigning Production Categories to Retail Chicken Parts

The goal of this study was to compare the fraction of *E. coli* resistant isolates to antibiotics among a variety of poultry products available in markets common in Southern California. We cultured *E. coli* isolates (*n* = 325) from wings, drumsticks, and chicken breast parts from several brands for testing resistance to seven antibiotics. We tested the hypothesis that there would be differences in antibiotic resistance among *Conventional*, *No Antibiotics* brands, and *Humane Family Owned* brands.

We cultured *E. coli* isolates (*n* = 325) from wings, drumsticks, and chicken breast from several brands and tested for resistance to seven antibiotics chosen for their use in chicken and importance to human health. All meat was purchased and processed in 2015, and all chicken labeling definitions are given to the best of our ability as they were regulated in 2015.

Seven different brands of retail chicken meat parts were evaluated and grouped into three main production categories—*Conventional*, *No Antibiotics*, and *Humane Family Owned*. *Conventional* chicken for our purposes is that which has no statement on antibiotic use. We define *No Antibiotic* chicken as any that has a USDA-approved statement on its packaging that is against the use of antibiotics. The *Humane Family Owned* chicken grouping includes only the brands purchased that both explicitly prioritize the well-being of the animal in a meaningful way and are third-party certified.

There are two different third-party groups that conducted the animal welfare certifications for the *Humane Family Owned* brands sampled in this work. These are application-based certifications earned through third-party inspection, and are not USDA-developed labels. The USDA does not have its own animal welfare specific label. The first is the Global Animal Partnership, which approves use of the label “Animal Welfare Certified” along with a step rating from 1 to 5+ that indicates how similar the animal’s environment was to a natural environment (with 5+ being the most similar). The other is Humane Farm Animal Care, which approves use of the label “Certified Humane.” If a farm is approved under either of the two standards, they can use the label on their packaging and consumers can easily access the meaning behind the label on the program’s respective website. Both the Global Animal Partnership and Humane Farm Animal Care conduct re-inspections every 12–15 months [41,42]. While both are respected certifications, the “Certified Humane” label from Humane Farm Animal Care label may be more recognizable to consumers because it has been used since 1998 while the “Animal Welfare Certified” label from the Global Animal Partnership was first used in 2008.

Initially we intended to compare *Conventional* versus *Organic* chicken in this study. The research into the exact details of the production categories and the purchasing of the chicken parts was done concurrently. It became apparent throughout the course of studying the production schemes that the *Organic* chicken actually fell into two different categories—*No Antibiotics* and *Humane Family Owned*. As a result of the division of what would have been the *Organic* production category, there are fewer isolates measured for antibiotic resistance for the *No Antibiotics* category than for either the *Conventional* or *Humane Family Owned* categories.

### 2.2. Meat Selection and Bacterial Purification

Raw chicken parts including drumsticks, wings, and breasts, were purchased in consumer markets in Los Angeles, California. Based on packaging labels, the chicken parts were grouped into three chicken production categories: *Conventional*, *No Antibiotics*, and *Humane Family Owned*.

After purchase, products were stored at 4 °C overnight until processing the following morning. Five 100-g samples were taken from each meat part purchased by aseptically removing the chicken from manufacturer packaging. Samples were transferred to individual, sterile Stomacher bags (VWR, Radnor, PA, USA, catalog number 11,216–902) and 125 mL of MacConkey broth was additionally added to the Stomacher bag [38]. To allow for sufficient dislodging of bacteria, the sealed bag was mixed at about 110 RPM (0.136× *g*) on an orbital platform shaker (Barnstead Thermolyne Roto Mix, 50,800) for five hours at 25 °C. Because we specifically aimed to minimize enrichment while allowing sufficient time for bacterial extraction, we did not use a 24 h contact time as some others have done. Immediately after shaking, 50 μL of the inoculated liquid was spread onto violet red bile agar (VRBA) plates in triplicate [29]. Plates were incubated at 35 °C for 24 h. After incubation, presumed *E. coli* colonies where chosen at random and streaked onto CHROMagar plates (Hardy Diagnostics, Santa Maria, CA, USA) for *E. coli* selection. *E.*
*coli* colonies were blue while other gram negative bacteria were colorless. A total of 75 confirmed *E. coli* isolates per meat brand were purified twice on VRBA plates and finally on CHROMagar plates. Each purification plate was incubated at 37 °C for 24 h. Isolated *E. coli* colonies (315–361) were randomly chosen for evaluation of antibiotic resistance as described below. The same isolates were used for all seven antibiotics.

### 2.3. Antibiotic Resistance Testing: Kirby-Bauer Disk Diffusion Susceptibility Test

Seven antibiotics representing a range of distinct antimicrobial classes and relevant to both livestock agriculture and human medicine were tested by the standard Kirby-Bauer disk diffusion susceptibility test in this work. Antibiotic resistance was evaluated against doxycycline, levofloxacin, ampicillin, cefoperazone, gentamicin, oxytetracycline, and erythromycin. Almost all disk diffusion cutoffs were defined by the CLSI (Clinical and Laboratory Standards Institute) [43] with the exception of erythromycin [44]. Because erythromycin is not traditionally used in developed areas for *E. coli* derived illnesses, it is not included in the CLSI, FDA, or EUCAST (European Committee for Antimicrobial Susceptibility Testing) standards.

In the disk diffusion method, each bacterial isolate (50 µL) was uniformly spread onto the surface of a Mueller-Hinton agar (MHA) plate with a sterile metal spreader to form an even film. Six-mm diameter antibiotic paper disks for levofloxacin (5 µg), doxycycline (30 µg), oxytetracycline (30 µg), gentamicin (10 µg), cefoperazone (75 µg), ampicillin (10 µg), and erythromycin (15 µg) (BD Diagnostic Systems) were placed on the surface of each seeded MHA plate using a pair of sterile forceps. Plates were incubated at 35 °C for 16–18 h. During incubation, antibiotic agents diffused outwards, creating regions of inhibition within the microbial lawn. All zones of inhibition were measured via either ruler or caliper. Using the Kirby-Bauer disk diffusion method, resistance to antibiotics was evaluated for over 300 chicken isolates per antibiotic.

### 2.4. Statistics and Data Analysis

Past work has grouped intermediate isolates with the resistant category citing being conservative on the side of consumer safety [35,45], while others have grouped them with susceptible. While both groupings are appropriate and acceptable, we chose to leave intermediate as its own grouping for the sake of being scientifically conservative.

The main purpose of this work was to understand whether there were significant differences between *Conventional*, *No Antibiotics*, and *Humane Family Owned* retail chicken meat. To determine whether there were significant differences between the production categories, following the standard convention, a chi-squared test was used with significance defined by *p* ≤ 0.05, unless expected frequencies were less than five, in which case a Fisher’s exact test was used with the same significance rule [46,47]. RStudio was used for all statistical analyses.

Standard errors for the fraction of resistant isolates of each brand were found from 95 percent confidence intervals of the sampling distribution of the proportion of isolates susceptible.

## 3. Results

### 3.1. Fraction of Resistant, Intermediate, and Susceptibility Isolates from Each Category of Chicken

For each category of chicken – *Conventional*, *No Antibiotics*, and *Humane Family Owned*, the fraction of isolates each that were susceptible, intermediate, or resistant to antibiotics were found. All samples from all companies were susceptible to levofloxacin, doxycycline, oxytetracycline, gentamicin, and cefoperazone.

Resistance was found to both ampicillin and erythromycin as seen below in Figure 1. Over half of the isolates were resistant to ampicillin in both *Conventional* (56.2%) and *No Antibiotics* (60.7%) chicken, and more than 90% of isolates were resistant to erythromycin in both *Conventional* (94.0%) and *No Antibiotics* (96.4%) chicken.

There was less resistance in the *Humane Family Owned* chicken isolates than the *Conventional* and *No Antibiotics* chicken. This relationship was similar for both ampicillin and erythromycin resistance. Significantly more isolates were susceptible to ampicillin when the chicken was produced by a *Humane Family Owned* company (55.7% susceptible) instead of a *Conventional* company (16.4% susceptible). Similarly, when the chicken was produced by a *Humane Family Owned* company, 8.7% of the isolates were susceptible to erythromycin, while only 1.3% were susceptible when the chicken was produced by a *Conventional* company.

Additionally, the percentage of *E. coli* isolates resistant to erythromycin was less when produced by a *Humane Family Owned* company (77.2% resistant) versus a *Conventional* company (94%). Because the CLSI standards for the Kirby-Bauer disk diffusion method include an intermediate breakpoint range, the percentage of resistant isolates is not simply 100% minus the percentage of susceptible isolates, which is why the differences in resistant isolates between the chicken brands are discussed separately from the differences in susceptible isolates. For resistance to either antibiotic, the fraction of isolates resistant, susceptible, or intermediate between *Conventional* and *No Antibiotics* chicken are similar.

### 3.2. Statistical Differences in Antimicrobial Resistance in Chicken Production Groupings.

There were no statistical differences between *Conventional* and *No Antibiotic* production, however there was a difference between these two groups and *Vertically Integrated* production, as illustrated in Figure 2. When all three production types were analyzed, there was significant association between the fraction of isolates susceptible and the corresponding production type (ampicillin *p* = 6.3 × 10^−15^; erythromycin *p* = 1.9 × 10^−4^), but when only the *Conventional* and *No Antibiotic* categories were analyzed together, there was no longer a significant association (ampicillin *p* = 0.5341, erythromycin *p* = 0.3248).

*Conventional* chicken had a fraction of 0.17 ± 0.05 *E. coli* isolates susceptible to ampicillin, while *Humane Family Owned* chicken had a greater fraction of 0.57 ± 0.05. Similarly, the fraction of *E. coli* isolates susceptible to erythromycin in *Conventional* chicken was 0.018 ± 0.012, and was 0.57 ± 0.07 for *Humane Family Owned* chicken.

To summarize, there was no statistical difference between the fraction of susceptible *E. coli* isolates produced through the *Conventional* compared to the *No Antibiotics* category, when considering sensitivity to either ampicillin or erythromycin. The *Humane Family Owned* category was, however, distinguishable from both of the other chicken production categories, with a significantly higher fraction of isolates susceptible to both ampicillin and erythromycin.

## 4. Discussion

In this work, *E. coli* isolates from *Humane Family Owned* poultry were significantly less resistant to ampicillin and erythromycin than those from either the *No Antibiotics* and *Conventional* categories of meat, suggesting that the fraction of antibiotic-resistant *E. coli* differs depending on the type of poultry production system. In this study, as among previous literature there was no significant statistical difference in the fraction of antibiotic-resistant *E. coli* among *Conventional* and *Humane Family Owned* isolates [35,45,48]. The meaningful and hopeful result from this work is that there *is*, however, a meaningful difference in the antibiotic resistance of *E. coli* isolates from farms with *Humane Family Owned* production.

### 4.1. Environmental and Human Health Relevance of Erythromycin Resistance in Gram Negative Isolates

As mentioned above in the methodology, there are currently no CLSI, EUCAST, or USFDA antibiotic susceptibility testing breakpoints standardizing *E. coli*’s susceptibility to erythromycin because it is not considered clinically relevant. While it is true that erythromycin is not used to target *E. coli* infections in humans, erythromycin susceptibility of *E. coli* has been studied in previous works [49,50,51], and there is a very well-documented connection between antibiotic resistance in humans and the environment [52,53,54,55,56]. Much of the reason for antibiotic resistance’s threat to public health is the particular proclivity of antibiotic resistance genes to be shared between bacteria. So long as the bacteria share, at least temporarily, a common habitat [57], and the fitness cost to the host cell is favorable [58], antibiotic resistance genes can be disseminated via conjugative horizontal gene transfer in dense microbial environments such as those in concentrated feeding animal operations (CAFOs). Of particular relevance to erythromycin resistance and *E*. *coli* is that horizontal gene transfer can take place beyond species boundaries, potentially even to pathogenic bacteria [58,59,60].

While the National Antimicrobial Resistance Monitoring System for Enteric Bacteria (NARMS) aids in the understanding of antibiotic resistance occurrence and proliferation in the United States, it is limited by only documenting “clinically relevant” antibiotic resistance. It was seen in our work that there was decreased antibiotic susceptibility in *Conventional* and *No Antibiotic* chicken compared to *Humane Family Owned*. Given the highly mobile nature of antibiotic resistance, it is important to document not only clinically relevant antibiotic resistance if the aim is to understand the spread, and not only the occurrence of antibiotic resistance.

### 4.2. Significantly FewerAntibiotic-Resistant Isolates in Humane Family Owned Chicken

It was found in this work that there were significantly fewer resistant isolates in chicken considered *Humane Family Owned* than *Conventional* or *No Antibiotics.*

Other previous literature has had conflicting results with respect to antibiotic resistance comparisons in various types of meat products. Bacteria on meat from organic poultry farms were shown to have lower antibiotic resistance compared to *Conventional* farms for *Campylobacter* [36], *E. coli* [11], and *Salmonella*. Miranda et al. (2016) [61] found that *Conventional* meat brands had higher odds of carrying antibiotic resistance than antibiotic-free chicken products. Similarly, Zhang et al. (2011) [37] found that *E. coli* as well as *Enterococcus* spp. on *Conventional* retail meat are more likely to be more resistant to some antibiotics than on samples with labels stating *No Antibiotics*.

Other studies, conversely, have found ARB resistance higher in organic products or having similar levels of ARB among meat products regardless of farming practice. Farming practice showed similar frequency of antibiotic resistance when *Conventional*, “Organic”, and “Raised Without Antibiotics” (RWA) poultry brands were compared in a study conducted by Millman et al. [35]. Similarly, Obeng et al. (2012) [39] found resistance to bacitracin, erythromycin, and tetracycline in most of the isolates collected from both *Conventional* and free range poultry and concluded that there was no significant difference in antibiotic resistance in *Enterococci* between both types of poultry farming. Saleha et al. (2009) [62] found antibiotic-resistant *E. coli* in day old chicken on commercial farms before introduction to any feed and water, therefore one possible explanation for the similar resistance levels in conventional and organic poultry could be due to contamination in the farming facilities.

While the *No Antibiotics* chicken brands do have more stringent antibiotic use standards than those of *Conventional* chicken, it was found that there was no difference between the fraction of resistant *E. coli* isolates between the brands. It has been hypothesized in other works that while animals may have been raised with no antibiotics, they are still often given antibiotics *in ovo* (injected into the egg) or within 24 h of the chick’s life. In our work, NA-2 did have a *Raised without Antibiotics* label, and this hypothesis aligns with what our results show. However, our other *No Antibiotics* brand name designation had a *No Antibiotics Ever* label, and therefore these chickens should not have received antibiotics at any point in their life or *in ovo*. Additionally, chicken labeled with a *No Antibiotics Ever* label only undergoes a one-time desk audit to ensure adequate segregation between animals given antibiotics and those that were not. However, there is not an in-person inspection, and given these results, questions may be raised regarding the amount of care taken to ensure that the segregation standards are met. Slaughter and packaging are often outsourced, making it difficult for a brand to reliably have much control over these production aspects. Even on-site (not mixed) slaughtering can play a role in antibiotic resistance. NARMS shows that there have been similar resistance rates in the USA in *E. coli* from slaughterhouses and retail meat [63], meaning that slaughtering practices may allow for the spread of antibiotic resistance in retail meats.

While there are only two brands in the *Humane Family Owned* category of our work, we would have liked to have examined more, but finding poultry raised to this standard of care was extremely difficult. Again, while the two brands were not evaluated by the same third party for humane raising, they were both more humanely raised than what the other standards required in our work. Similarities between the two third party certifications include that there were no antibiotics or antibiotic by-products in animal feed, there was a specific standard of little stress and discomfort during the slaughtering process, there was a standard for the chickens to be able to live their life in a natural way, and a standard of natural and humane raising was explicitly stated in their requirements. Both brands were also family-owned, and the same production company was responsible for every aspect of the animal’s life, instead of outsourcing to other companies as is common practice. While one of the brands purchased chickens from farmers, they only did so from small, locally owned farms with whom they had a good and transparent relationship.

It should be made clear that the discussion for this work’s purposes around the humane raising of chickens refers to the prioritization of the health of the animal, not to any specific “humane” third party standard. From our research, we are confident that both brands in the *Humane Family Owned* production category prioritize the health of the animal in ways significantly beyond those of the *No Antibiotics* brands.

### 4.3. Support for the Prioritization of One Health Standards to Combat Antimicrobial Resistance

The One Health standard is a unique approach because it promotes the welfare of humans, animals, and the environment as collaborative priorities that result in the health of all three. This is different than the *No Antibiotics* in poultry production approach to human health, mainly because the health of animals in a One Health approach is an end in and of itself and does not serve only to better the health of humans.

In this study, one brand each from the *No Antibiotics* and *Humane Family Owned* production categories, NA-2, and VI-1, are USDA certified organic. While the certified organic label ensures a certain quality standard during the life of the chicken, such as “accommodating” their behavior, providing organic feed, and no hormones or antibiotics after 24 h of life [64], these standards are ultimately made to benefit the health of humans, not the well-being of the animal.

There was a significant difference between the *No Antibiotics* and *Humane Family Owned* brands overall, both of which included organic brands. The USDA organic label, despite its restrictions on antibiotic usage, resulted in no difference between the *Conventional* and *No Antibiotics* production categories overall. The main differences between the *Humane Family Owned* chicken and the others was that the companies were vertically integrated and therefore had more oversight, and they followed a strict set of added requirements to specifically characterize the chicken as humanely raised.

## 5. Conclusions

An interesting result of our work is that in fact when prioritizing the health of the animals themselves, fewer antibiotic-resistant *E. coli* isolates were found in the chicken. This work and previous literature show that retail chicken meat cannot easily be categorized due to various factors including sharing of slaughterhouse facilities and possible antibiotic use during the first 24 h of life, which is allowable under the designation of “Organic”. While limiting in scope, our findings suggest that *No Antibiotics* chicken, while raised differently than *Conventional* chicken, does not actually result in a difference in the frequency of antibiotic-resistant *E. coli* isolates for the consumer in retail chicken meat, and is thus misleading. This result has similarly been found in other works. The difference and positive contribution of this work is that there was a significant difference in the *Humane Family Owned* chicken brands, which took extra measures to prioritize the health of the chicken. While these brands are not as commercially available as others, they are a better choice for the consumer who is specifically interested in minimizing the amount of antibiotic resistance in their raw chicken meat. The WHO has already declared the implementation of the One Health in the combat of antimicrobial resistance, and the findings of our work show that there are clear benefits to following the WHO’s guideline.

## Figures and Tables

**Figure 1 animals-10-02217-f001:**
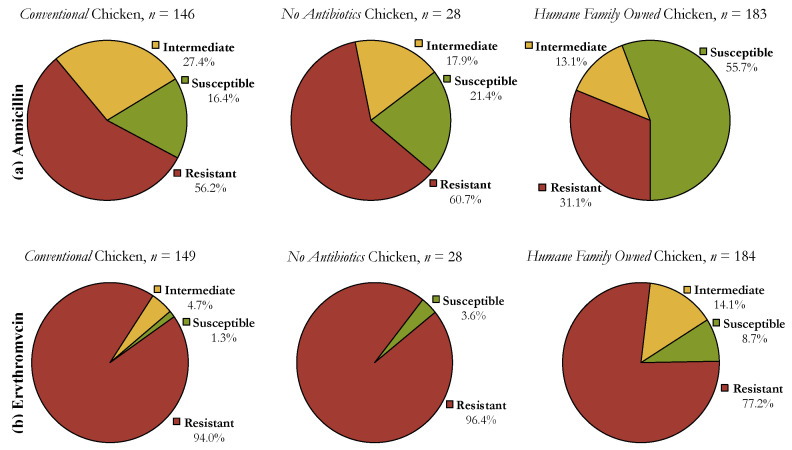
Pie charts showing the percentage of resistant, intermediate, and susceptible isolates for each of the three production categories of chicken—*Conventional*, *No Antibiotics*, and *Humane Family Owned*. (**a**,**b**) demonstrate resistance and susceptibility to ampicillin and erythromycin, respectively.

**Figure 2 animals-10-02217-f002:**
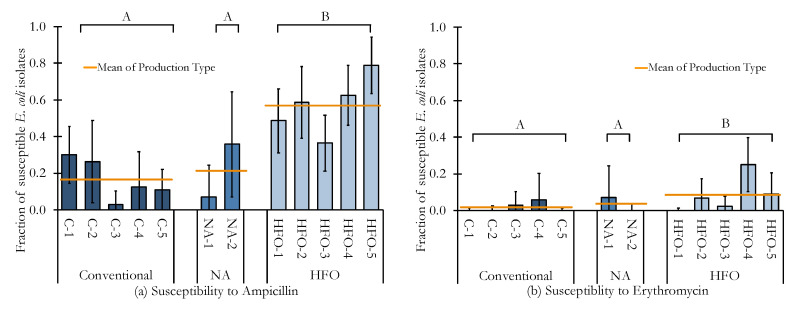
Each column per production grouping represents a unique retail broiler chicken option. Error bars represent 95% confidence intervals of the sampling distribution of the proportion of isolates susceptible. Error bars that would theoretically extend past zero were truncated. Statistical significance letters A and B shown on the graphs refer to whether the fraction of isolates susceptible are different between production groupings. Same letters show no significant difference, while different letters show a difference between groupings. The vertical orange line represents the mean fraction of susceptible *E. coli* isolates to (**a**) ampicillin and (**b**) erythromycin grouped by *Conventional*, *No Antibiotics (NA)*, and *Humane Family Owned (HFO)*.

**Table 1 animals-10-02217-t001:** Package labeling descriptions of antibiotic use and humane standards on the retail broiler chicken in this study ^1,2^.

	Certified Humane	GAP Level 3+	Organic	No Antibiotics Ever	Raised without Antibiotics	Free Range	All Natural	Conventional
Antibiotic Use ^3^	Only when sick and under veterinarian care After treatment, the label *may* still be used ^4,5^	Only when sick and under veterinarian care After treatment, the label *may not* be used (5) ^6^	May be injected *in ovo* ^7^ and used on first day of life Prohibited after first day of life Sick animals may be treated with antibiotics under veterinary supervision but the label *may not* be used ^8^	Only when sick and under veterinarian care After treatment, the label *may not* be used (5)	May be injected *in ovo* Prohibited after birth Sick animals may be treated with antibiotics under veterinary supervision but the label *may not* be used ^9^	No Requirement	No Requirement	No Requirement
Outdoor Access	No Requirement ^10^	Outdoor access must be at least 25% of indoor area Outdoor access required beginning at four weeks of life (6)	Specific amount of required outdoor space is not given ^11,12^ Outdoor access must be available for at least 50% of life.	No Requirement	No Requirement	No Requirement	No Requirement	No Requirement
Animal Byproducts in Feed	Allowed	Prohibited	Prohibited	Allowed	Allowed	Allowed	Allowed	Allowed
Space Requirements	1 sq. ft per 6 lbs of animal (4)	1 sq. ft per 6.5 lbs of animal (4)	No Requirement	No Requirement	No Requirement	No Requirement	No Requirement	No Requirement
GMOs in Feed	Allowed	Allowed	Prohibited	Allowed	Allowed	Allowed	Allowed	Allowed
Light Requirement	Six hours of continuous darkness per day after first day of life (4)	Eight hours of continuous darkness per day after first day of life (6)	No Requirement	No Requirement	No Requirement	No Requirement	No Requirement	No Requirement
Slaughter Requirements	Birds must be stunned and insensible to pain Slaughter facilities undergo annual audits (4)	Only standard USDA slaughter requirements ^13^	Only standard USDA slaughter requirements	One-time desk audit to ensure adequate segregation between animals given antibiotics and those that were not ^14^	One-time desk audit to ensure adequate segregation between animals given antibiotics and those that were not ^15^	Only standard USDA slaughter requirements	Only standard USDA slaughter requirements	Only standard USDA slaughter requirements
Audit by a Third Party	n/a—They are a third party that performs farm audits	n/a—They are a third party that performs farm audits	Yes ^16^	None	None	None	None	None
Clearly Published Standards for Label Use	Yes (4)	Yes (6)	Yes ^17^	No, farmers submit documentation to the USDA to apply for this label (16)	No, farmers submit documentation to the USDA to apply for this label (9)	No, farmers submit documentation to the USDA to apply for this label ^18^	Yes	n/a

^1^ To the best of the authors ability, this table reflects labeling requirements as established when the experimental work was conducted (2014). ^2^ References are only excluded in the following cases: (1) Cases where the information came directly from the chicken retailer (in order to protect anonymity), and (2) Cases of *No Requirement* were determined by conducting a thorough exploration of published standards. ^3^ As a result of the full implementation of FDA’s Guidance #213 in 2017 (US Food and Drug Administration, 2013), “medically important antibiotics” are no longer allowed in animal feed for growth promotion or disease prevention purposes. This work was conducted before these regulations were formalized in 2017, Regardless, there is much speculation as to whether or not the use of “non medically important antibiotics” can be regulated in cases where there are no third-party audits. Even more, as discusses further below, there is reason to believe that the use of “non medically important” antibiotics could still create issues for human health as a result of horizontal gene transfer. ^4^ (Humane Farm Animal Care, 2014). ^5^ (United States Department of Agriculture, 2020). ^6^ (Global Animal Partnership, 2018). ^7^
*In ovo* refers to antibiotics being injected into the egg before the chick is hatched. ^8^ (ATTRA Sustainable Agriculture & USDA, 2015). ^9^ (USDA Food Safety and Inspection Service, 2019). ^10^ The *Certified Humane* standards state that this is because some chickens only live 5–7 weeks. Because they do not go outside until they have feathers (4 weeks), some flocks would never get to go outdoors because of weather. (Humane Farm Animal Care, 2015). ^11^ The length of time the birds are required to have access at one time is not given. Because there are many specifics missing from this claim, the animal conditions can vary widely between facilities. ^12^ (United States Department of Agriculture Food Safety and Inspection Service, 2015). ^13^ However, does use Controlled Atmosphere Stunning (CAS). Controlled atmosphere (gas) systems are increasing in use and account for the majority of poultry slaughter in the UK. Advantages include consistency in application across all birds in the system and, since these are stun-kill systems, there is no risk of the birds recovering consciousness during bleeding. Another significant advantage is that the birds can remain in the transport modules throughout the process, avoiding the need for additional live handling. However, there is debate regarding the humaneness of this technology. ^14^ (Greener Choices, 2020). ^15^ (Greener Choices, 2020). ^16^ (USDA, 2017). ^17^ (USDA Organic, 2014). ^18^ (USDA Food Safety and Inspection Service, 2019).

**Table 2 animals-10-02217-t002:** Package labeling descriptions of antibiotic use and humane standards on the retail broiler chicken in this study ^1^.

		Certified Humane	GAP Level 3+	Organic	No Antibiotics Ever	Raised without Antibiotics	Free Range	All Natural
Conventional	C-1	☐	☐	☐	☐	☐	☐	■
	C-2	☐	☐	☐	☐	☐	☐	■
	C-3	☐	☐	☐	☐	☐	☐	■
	C-4	☐	☐	☐	☐	☐	☐	■
	C-5	☐	☐	☐	☐	☐	☐	■
No Anti-biotics	NA-1	☐	☐	☐	■	☐	☐	■
	NA-2	☐	☐	■	☐	■	■	■
Humane Family Owned	HFO-1	☐	■	■	☐	☐	■	☐
	HFO-2	☐	■	☐	■	☐	☐	☐
	HFO-3	☐	■	☐	■	☐	☐	☐
	HFO-4	■	☐	☐	☐	☐	☐	■
	HFO-5	■	☐	☐	☐	☐	☐	■

^1^ Filled (black) squares indicate that the label was present on the retail chicken packaging, while unfilled (white) squares indicate that the label was not present.

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
