# Peer review of "Antibiotic Resistance of Escherichia coli Isolated from Conventional, No Antibiotics, and Humane Family Owned Retail Broiler Chicken Meat"

_animals, 2020, doi:10.3390/ani10122217_

Round 1

Reviewer 1 Report

The manuscript is much improved and almost ready for publication.  A few remaining points of clarification:

1 – Lines 120-131:  I am still confused by the difference in the various “no antibiotic” labels and the USDA seal.  Are the labels themselves not regulated, while the seal is granted after an application to the USDA and the application itself is not public?  And the seal in conjunction with the (self administered) label guarantees that the USDA reviewed and approved an application.  Is this “seal” the same as the “Organic” label described in the above paragraph?  Or is the “seal” a separate certification?

2 – The numbers in results still do not add up properly.  A total of 75 isolates were isolated.    You then specify a subset of these were chosen for antibiotic resistance. 

  • Was this 75 isolates per farm?
  • How were the subsets chosen?
  • Were the same isolates used for all 7 antibiotics?
  • How did you get the number 325 (line 253)?
  • Why were different numbers of colonies tested (357 for ampicillin; 361 for erythromycin)? If some were lost to plate damage, just state that between x and y colonies were tested per antibiotic.
  • The presentation of the Humane Family owned susceptibility lines 277-283 was confusing. The phrase “increased be 39.3%” occurs is a vacuum (I cannot figure out which numbers you are comparing).  Are these absolute numbers or percentage increases / decreases? I would be more specific (55.7% susceptible versus 21.4% and 16.4%) to avoid confusion.  Same with “16.8% fewer isolates”

3 – In the discussion, a great deal of emphasis seems to be focused on “animal centric” farming practices in vertical operations while the importance of slaughtering practices seemed downplayed.  What do you think was the relative contribution to on-site (therefore not mixed) slaughtering to antibiotic resistance?

Minor edits:

Line 88:  ” Agricultural 89 antibiotic standards from the USDA have changed since, but in 2014, and the agricultural use of 90 “medically important” antibiotics decreased by 36% from 2014 to 2018.” Should read “Agricultural 89 antibiotic standards from the USDA have changed since 2014, and the agricultural use of 90 “medically important” antibiotics decreased by 36% from 2014 to 2018.

Line 363: is this difference due to exposure at farming facilities or hatching facilities?  Could the results also be caused by antibiotic treatment before 24 hours?

Line 374: typo “ . . . and those that were not

Reviewer 2 Report

The term “Humane  Family Owned” is confusing. There is “CERTIFIED HUMANE” certification, but apparently authors includes not only chickens with this certification in this category.

I suggest to keep title as: Antibiotic Resistance in Escherichia coli Isolated  from Conventional, No Antibiotics, and “Family Owned” or “Family Farmed”  Retail Broiler Chicken Meat.

Anyway, the “Family Owned” concept should be explained, in order to make clear what kind of productions includes.

The objective of the present study is not clear and should be state at the end of the introduction. Is stated at the beginning of MM, line 195 : “The goal of this study was to compare the fraction of E. coli resistant isolates to antibiotics among a variety of poultry products available in markets common in Southern California”. However, it should be further described in the introduction.

The three categories included in the study (table 2), should be also indicated in table 1, in order to allow a better understanding of which package labeling description correspond under each studied category.

Manuscript is well written , and results are very interesting, however , the categories of the sampled retail chicken meats taken should be clarified, as well as the criteria of the selection of the antibiotics tested.

Author Response

This manuscript is a resubmission of an earlier submission. The following is a list of the peer review reports and author responses from that submission.

Round 1

Reviewer 1 Report

The study design is interesting and compelling, while being pertinent to both human health and poultry production.  Unfortunately, the manuscript itself is confusingly written which distracts from the actual results.  I would suggest a complete rewrite focusing on a more logical organization to the introduction and conclusions as well as moving some of the information included in the conclusion to the Materials and Methods section (where the information makes more sense).  As far as the study itself, some of the numbers for isolates tested don’t make sense with the study design.  since you have the data and discuss antibiotic resistance in the discussion, consider comparing the organic farms to the non-organic in addition to the other statistics.

I would encourage you to resubmit this paper after a re-write because the results are interesting and relevant.

I have listed some specific issues, but all sections need to be better organized:

“Conventional,” “Non Antibiotics,” and “Vertically Integrated” are not consistently put in italics.

Line 33 – “We grouped retail chicken from Los Angeles into three production groups . . . “ This sentence sounds like the actual chickens were grouped into categories as opposed to the companies.  Better would be “Companies selling chicken products in LA were grouped . . . “

Line 42 – You do not actually state what the statistical difference was: more or less bacterial resistance?

Line 43 – You do not need to justify your choice of bacteria within the abstract

Lines 54-55 The sentence is unclear as to what is being connected: the amount of chicken and antibiotic resistance?  Deaths occurring 2050?  I would make these two points separate sentences for clarity: 1) connection between the amount of chicken produced and antibiotic resistance and 2) these resistant bacteria will result in one death every 3 seconds by 2050

Lines 61-62 Need a reference for the biome impact

Lines 68-69 “Infections caused by antibiotic resistant . . .” This sentence fits better with the first Introduction paragraph when discussing antibiotic resistance and human health.

Line 82 – “ . . . neighboring families, had . . “ this should be in present tense “neighboring families, have

Line 84 - “may also be affected” While not incorrect, not general English usage.  “may also be exposed

Lines 102 – what USDA seal?   

There are 3-4 different explanations of the categories you used (introduction, M+M and discussion).  I would consolidate the descriptions into the Introduction and remove from the Discussion.  The more succinct of these version is in 158-162 is fine for the M+M section.  I would also suggest a table comparing the retail companies based on antibiotic use, organic status, growing practices and processing facilities.

Line 138 – Define organic

Lines 157 and 164 – what kind of chicken product was purchased? Whole chicken, parts, etc? 

Line 157 – “were measured” is not correct English usage here.  I think you mean that the production practices of the companies were evaluated

Line 166 – So 5 drumsticks, 5 wings and 5 breasts were samples for each company for a total of 15 samples per company?  Then these 15 samples were spread on 3 plates each? Wouldn’t that be 45 plates per company?  Were the samples pooled at some point? I was not able to follow how many samples were collected and then plated.

Line 200 – So the “uncertain designation” is given with there was variability between samples?  Or when the diffusion may not have reached the susceptible zone?  If it is not clear the susceptibility is in a clinically relevant dose?  All the above?

Figure 1 – Why is there such a big discrepancy in the number of samples?  The figure legend is cut off in the PDF so I cannot read the full label.

Line 216-217 – The opening sentence really belongs in the Material and Methods sections.  Consider using this explanation in places of Lines 197-201

Line 220 -  For clarity, I would state that “all samples from all companies were susceptible to (antibiotic names here)”

Lines 223 “fraction of isolates” This sentence is hard to follow.  Are you saying that there were more susceptible isolates in the Vertically integrated farms?

Line 226-227 “The fraction of isolates resistant to erythromycin decreased by 16.8%” This sentence is awkward as well “16.8% fewer isolates were resistant to erythromycin in Vertically integrated farms.”

Line 228 – The paragraph would make more logical sense if you started with the results from the Conventional and No Antibiotic farms, then compared to Vertically Integrated.

Line 257-259 – Not necessary to restate M+M

Lines 260-266  It would be clearer to just state “There were not statistical differences between Conventional and No Antibiotic farms, however there was a difference between these two groups and Vertically Integrated.

Line 267-272 – Similar to above, the sentence structure is confusing.  Numbers of susceptible isolates should go into section 3.1 when discussing percentage changes in susceptibility.

Figure 2 – This is an excellent presentation, but I would replace the A and B bars with asterisk marks for significance (a more standard presentation)

Lines 288-294 – Nice summary

Lines 297 – need to define these anacronyms

Lines 312-319 This section would fit better into the Introduction when explaining the rational for the study

Lines 314-335 and Lines 394-409 Since you discuss organic, and know which companies are labelled as such, consider running statistics on organic / non-organic from your study

Line 345 – RWA anacronym stands for?

Lines 350-356 – excellent explanation

Lines 358-374 – Maybe include in M+M?  The relationship between these labels and your study is unclear.

Line 403 – I don’t understand the phrase “centric around health of humans” – do you mean that protecting human health is the goal and not the well-being of the animal?

Reviewer 2 Report

Line 21: there was no significant difference (p ≤ 0.05)

Did you mean P > 0.05?

Line 24-25: To our knowledge, this is the first time E. coli from Vertically Integrated chicken has been studied for antibiotic resistance.

There are a lot of other studies which tested AMR in E. coli from vertically integrated chicken.

Line 34: Vertically Integrated is not a federally regulated category in the United States

Vertically integrated poultry production is federally regulated by USDA-FSIS

Line 40: no significant difference (p ≤ 0.05)

Did you mean P > 0.05?

Lines 44-48: To our knowledge, Vertically Integrated consumer chicken has not previously been studied for its antibiotic resistance. This work contributes to a better understanding of a potential strategy of chicken production for the overall benefit of human health, giving evidentiary support to the One Health approach implemented by the World Health Organization.

Majority of the chicken produced in the US is through Vertically Integrated production. The system is federally inspected by the USDA-FSIS. This is not a new strategy; it has been there from 1950s.  Plenty of AMR work has been done and published on vertically integrated chicken. Also, what do you mean by consumer chicken? It might be illogical to propose an already existing system as a potential new strategy.

Lines 66-68: While it is no longer allowed in the United States to give antibiotics to food animals intentionally as growth promoters, they are still allowed for feed efficiency and the prevention of disease,

The authors say that antibiotics are not allowed as growth promoters, but then say they can be used for improving feed efficiency. Are these two applications different?

Lines 71-72: While much attention has been focused on the clinical overuse of antibiotics, up to 70 percent of antibiotics produced in the United States in 2008 were sold for use in livestock alone [9].

This is questionable. The reference is not sufficient to prove the credibility of this statement. You need to provide proper reference when you make such a strong statement.

Lines 85-87: Once antibiotic resistant bacteria become a part of the human gut environment, it is known that antibiotic resistance genes are able to move horizontally, especially through conjugative gene transfer, to other bacteria.

Need reference.

Lines 108-119: For our purposes, we assign three different categories based on third party certification and statements on the retail labels: Conventional, No Antibiotics and Vertically Integrated. The descriptions of the three are further explained below, but briefly, the Conventional chicken tested was known to be treated with antibiotics sub-therapeutically at the time of the study. The No Antibiotics category includes brands which make reference to no antibiotic usage but may share facilities for slaughter with brands employing antibiotics. Vertically Integrated poultry prohibits sub‐therapeutic use of  antibiotics, ionophores, beta agonists, and/or sulfa drugs before and after hatching, provides vertically integrated production (no sharing of slaughter facilities, for example) and is third-party certified by a corporation prioritizing the well-being of the animal, either by the Global Animal Partnership or Certified Humane standards in our work. Vertically Integrated indicates that the chicken is managed from birth through slaughter by the same company without sharing of facilities. While it is not a production category of poultry, it is an appropriate descriptor of the production process of these brands in our work.

The distinction between the categories are not clear. Was the conventional chicken from a vertically integrated company? How did you know the ‘No antibiotics’ category may share facilities for slaughter with brands employing antibiotics?

Majority of the vertically integrated companies have moved to No Antibiotics Ever (NAE) system. How did you differentiate these two?

Lines 136-139: It is important to note that while the antibiotic use in poultry labeling and production is not easily discernable from labels, and even often misleading, there are still many environmental and health benefits to consuming organic chicken products outside of the antibiotic resistance health effect.

Where did you get this information from? What are the additional health benefits of consuming organic chicken over non-organic chicken? This manuscript is not about organic chicken, then why are you explaining the benefits of organic chicken here?

Introduction Lines 53-150: Why did you choose E. coli in this study? Why not other potential pathogens present in chicken?

Lines 148-149: from several brands for testing resistance to seven antibiotics.

What was the criteria for choosing 7 antibiotics while there are 11 classes of antibiotics in CLSI?

Lines 157-162: Seven different brands of retail chicken meat were measured and grouped into three main production categories – Conventional, No Antibiotics, and Vertically Integrated. Conventional chicken for our purposes is that which has no statement on antibiotic use. We define No Antibiotic chicken as any that has a USDA-approved statement on its packaging that is against the use of antibiotics. The Vertically Integrated chicken grouping includes only the brands purchased that both explicitly prioritize the well-being of the animal in a meaningful way and are third-party certified.

Were the first two categories produced through vertically integrated system? There is a high chance they were. This is a major drawback of this study.

Lines 164-181: Please explain the colony morphology used for E. coli selection.

Line 191: plate with an antiseptic metal spreader to form an even film.

What is an antiseptic metal spreader? Did you mean sterile metal spreader?

Results: Lines 217-231: The wordings are confusing. Please provide P values for each individual result. Replace ‘percent’ by ‘%’.

What was the prevalence of E. coli in each category of meat? How many samples were positive out of the total samples tested?

No P values are given in Figure 1. Are these numbers different? If yes, please provide the actual P values.

There is a huge variation in the number of isolates you used for statistically comparing each category. How would you explain this?

Lines 262-266: When only considering  Conventional and No Antibiotic categories, however, there was no longer a significant association, showing that there was no meaningful difference in susceptibility between these two categories was no significant association between the susceptibility of E. coli isolates in the Conventional and No Antibiotic categories (ampicillin p = 0.5341, erythromycin p = 0.3248).

Please rewrite this sentence. Poor sentence structure.

Results: Lines 217-278: Why don’t you mention your values with stats rather than separating into two separate sections?

Figure 2: What are the labels on X axis? VI-1B, VI-2D etc.?